# Production and Tribological Characterization of Advanced Open-Cell AlSi10Mg-Al$_2$O$_3$ Composites

**Mihail Kolev \*** , **Ludmil Drenchev** , **Veselin Petkov and Rositza Dimitrova**

Bulgarian Academy of Sciences, Institute of Metal Science,
Equipment and Technologies with Center for Hydro- and Aerodynamics "Acad. A. Balevski",
Boulevard "Shipchenski Prohod" 67, 1574 Sofia, Bulgaria
\* Correspondence: mihail1kolev@gmail.com

**Abstract:** In this study, advanced open-cell porous AlSi10Mg-Al$_2$O$_3$ composites have been successfully fabricated by replication of NaCl space holders. The tribological behavior under dry sliding conditions at room temperature of composites with different pore sizes was studied via the pin-on-disk method, and wear parameters, such as the coefficient of friction (COF) and mass wear, were determined. Micro-hardness tests have been performed to investigate the change in mechanical properties after the processing of the composite materials. Microstructural observation was conducted by means of light microscopy and scanning electron microscopy (SEM) along with chemical micro-analysis using an X-ray energy-dispersive spectroscopy (EDS) system. The obtained results revealed that the investigated AlSi10Mg-Al$_2$O$_3$ composites possess lower COF and mass wear than the open-cell porous AlSi10Mg material when subjected to the same test conditions. Furthermore, it was also reported that the effect of pore size is insignificant to the COF, and in relation to mass wear, the composite material with the larger pores shows better results.

**Keywords:** AlSi10Mg-Al$_2$O$_3$; aluminum metal matrix composites; tribological characterization; mass wear; coefficient of friction





## 1. Introduction

The production of high-strength and light-weight composites has had a significant environmentally friendly effect by reducing emissions and improving the tribological and mechanical properties of manufactured goods intended for use in transportation and industrial machinery. Interconnected (open-cell) aluminum metal matrix composites (AMMCs) have been extensively studied by a large number of researchers and serve as a vital component of many functional and structural engineering applications, such as the development of wear-resistant and light-weight cylinder liners from aluminum alloy reinforced with graphite, and the production of energy-saving automotive brake rotors from aluminum reinforced with SiC [1–5].

One of the most extensively employed processes for obtaining AMMCs is the replication method. This method is characterized by its lack of sophistication in the experimental part and high extent of control over the shape, size and pore distribution of final cellular metal matrix structure [6]. The replication method is based on the fabrication of a preform, which might consist of different materials called space holders, such as NaCl [7–14], magnesium sulfate [15], carbamide [16], magnesium [17], saccharose [18], Acrowax [19], ammonium bicarbonate [20] and potassium carbonate [21].

To provide improved wear resistance and hardness to the softer matrix of the AMMCs, various types of reinforcing phases (RP) are introduced. Some of the most frequently used as RP materials, having excellent tribological behavior, are ceramic materials such as alumina (Al$_2$O$_3$) [22–26], silicon carbide (SiC) [22,27–30], graphite (Gr) [31], boron carbide (B$_4$C) [32] and titanium carbide (TiC) [33,34]. Due to its high thermal stability, good corrosion and

excellent wear resistance, alumina can be considered a convenient option for reinforcement of AMMCs [35].

Thanks to its cost-effective application and benefit of effective distribution of the reinforcement, liquid-state processing is a wide-spread route for the fabrication of AMMCs [36–38].

The composite material's wear behavior is influenced by the technological parameters of AMMCs fabrication, inasmuch as strong bonding between the metal matrix and the ceramic particles during the friction results in high wear resistance [39]. The tribological behavior is also influenced by the test parameters such as applied load and reinforcement size. An investigation of wear behavior at dry sliding conditions of $Al_2O_3/AA6061$ composites concluded that the composites showed improved wear resistance compared to the base alloy, and when the applied load was increased up to values above 230 N, a serious wear occurred in both the composite and base alloy [40].

A study of the effect of particle size on the wear behavior under dry sliding conditions of $AA7075/Al_2O_3$ (with 5 wt. % particles with sizes varying between 0.3 and 15 μm) composites obtained via powder metallurgy reported that the sample with the largest reinforcement size showed the finest wear resistance. The authors of the study also conducted an analysis of variance and determined that the most significant parameter influencing volume loss is the applied load [35].

Faiz Ahmad and others [41] fabricated $AA242/Al_2O_3$ (30 vol. %) through a squeeze casting method and conducted dry wear tests; their results indicated that with the increase in the wear load to 100 N, the results indicated an increase in the COF and a wear loss decrease in the composite when compared with the base material.

M. M. A. Baig and others [42] investigated the wear behavior at dry sliding conditions of $AA6061/Al_2O_3$ and compared it with a gray cast iron brake rotor. The authors reported lower wear rate results in the composite at lower brake power intensity, and even lower wear results in the composite at higher brake power intensity due to the better heat dissipation in the composite.

The authors in [43] obtained $AA\ 1100/Al_2O_3$ (3, 6, 9 and 12 wt. %) composites through a stir casting method and reported that the optimal results for mass wear and the maximum hardness were reached at the addition of 12 wt. % of reinforcement.

The aim of the current research is the fabrication of an advanced open-cell AlSi10Mg-$Al_2O_3$ composite material and the investigation of the effect of tribological interaction parameters, such as mass wear and COF, under dry sliding conditions. Furthermore, structural and chemical characterization of the composite material is to be performed.

Open-cell AlSi10Mg-$Al_2O_3$ composites were obtained through a liquid-state processing route by applying a replication method to produce the preliminary preform in which NaCl was used as a space holder material. To attain the desired levels of porosity in the composite and to effectively remove the space holder, a replication method was employed. The combination of the low-density, fatigue strength, high load-carrying capacity, thermal conductivity, excellent corrosion resistance and overall low-price characteristics of the aluminum alloy with the combination of the reinforcement $Al_2O_3$ particles with excellent wear and hardness behavior allows the AMMCs to be integrated in industrial processes for the manufacture of sliding contact bearings.

This study is a continuation of our previous studies [7,9,25,26] where open-cell aluminum-based composite skeletons were fabricated using the replication method and thereafter the skeletons were infiltrated by babbitt alloy.

## 2. Materials and Methods

### 2.1. Production Method and Materials

The advanced open-cell composite material consists of a high-porosity AlSi10Mg-$Al_2O_3$ composite skeleton. To attain the desired porosity, a replication method was used for the preparation of the skeleton. The production process of the preliminary preform starts with the preparation of a NaCl as a leachable space holder. The space holder particles were mixed with 6 wt. % water and 5 wt. % $Al_2O_3$ particles, after which the mix was

homogenized in a 3d powder blender—WAB T2F Turbula Heavy-Duty Shaker-Mixer (Willy A. Bachofen AG, Muttenz, Switzerland). The obtained mixture was compacted at a pressure of 1.5 MPa into a cylinder-shaped steel container. The moisture removal of the obtained "green" compacts was performed via drying them in a furnace at 200 °C for 2 h. The green dried compacts were sintered in a furnace at 800 °C ± 1 °C for 1 h, and the cooling of the obtained salt leachable preform was performed at room temperature. The next process step is the infiltration of the salt preforms with molten AlSi10Mg alloy. The preforms were preheated before fixation in a die at 680 °C ± 2 °C. The infiltration was conducted by employing the squeeze casting method with an applied pressure of 80 MPa for 60 s. The obtained composite material was cooled down at room temperature and the space holder was removed via dissolution in hot (70 °C) distilled water using an ultrasonic device.

The scheme in Figure 1 presents the main stages of the fabrication process of the high-porosity AlSi10Mg-Al$_2$O$_3$ skeleton.

## Obtaining NaCl preform

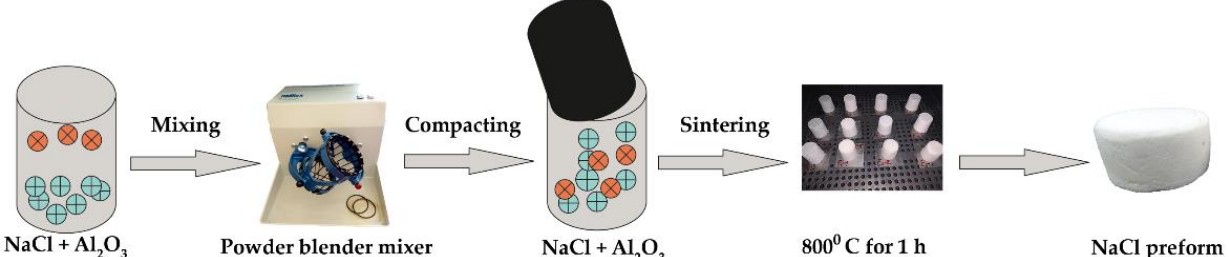

## Fabrication of AlSi10Mg + Al$_2$O$_3$ skeleton

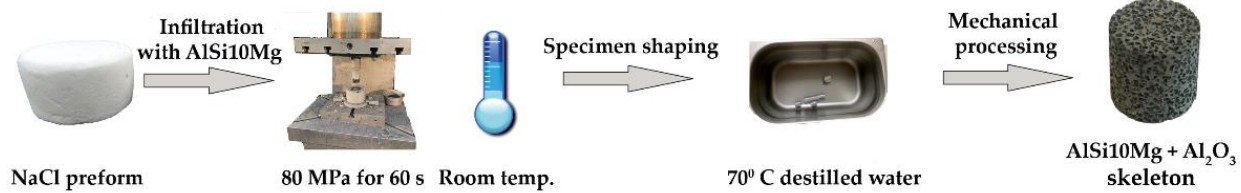

**Figure 1.** Fabrication process of high-porosity AlSi10Mg-Al$_2$O$_3$ skeleton.

The chemical composition of the base alloy used for the infiltration of the salt preform and the production of the AlSi10Mg-Al$_2$O$_3$ skeleton is given in Table 1. This specific alloy was chosen because of its low weight, post-processing flexibility and very good thermal and mechanical behavior. The reinforcement is achieved by Al$_2$O$_3$ particles with sizes varying between 300 and 400 μm (Figure 2a). Two types of composite skeletons with different pore sizes were produced. The pore size control was implemented using sodium chloride particles of two sizes: (1) 800 ÷ 1000 μm and (2) 1000 ÷ 1200 μm (Figure 2b). The porosity of composite with size (1) is 64%, while the porosity of composite with size (2) is 68%.

**Table 1.** AlSi10Mg alloy composition.

| Element | Si | Fe | Cu | Mn | Mg | Ni | Zn | Pb | Sn | Ti | Al |
|---|---|---|---|---|---|---|---|---|---|---|---|
| Concentration, wt. % | 9.0–11.0 | 0.55 | 0.05 | 0.45 | 0.2–0.45 | 0.05 | 0.10 | 0.05 | 0.05 | 0.15 | rest |

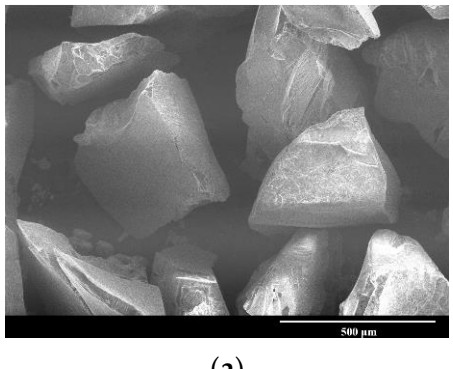
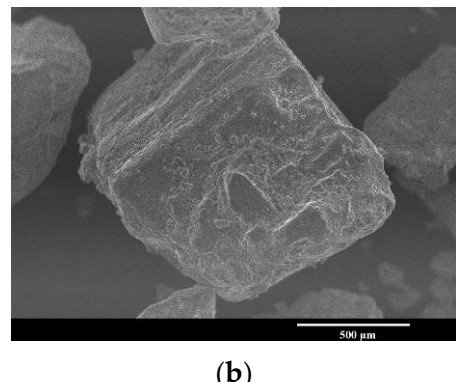

(**a**)                                                          (**b**)

**Figure 2.** SEM images of: (**a**) reinforcing phase (Al$_2$O$_3$) and (**b**) space-holder phase (NaCl).

### 2.2. Characterization Methods

Structural observation was performed via a light microscope, model Polyvar Met optical system (Reichert Jung, Wien, Austria) and a scanning electron microscope (SEM), model SH-5500P (Hirox Japan Co Ltd., Tokyo, Japan). Chemical micro-analysis was executed via an X-ray energy dispersive spectroscopy (EDS) system, model QUANTAX 100 Advanced (BRUKER, Kontich, Belgium).

Average Vickers hardness (HV) tests were conducted using a Micro-Duromat 5000 computer control semi-automatic micro Vickers hardness tester (Reichert Jung, Wien, Austria).

The wear properties of all test specimens were defined with a Ducom Rotary (Pin/Ball-on-Disk) tribometer, model TR-20 Ducom (Ducom Instruments Pvt. Ltd., Bangalore, India). The pin-on-disk installation was used to conduct dry wear tests at room temperature on test specimens (pins with a spherical tip) that were 10 mm in diameter and 20 mm high with the following test parameters: load 50 N, linear velocity 1.0 m·s$^{-1}$ and sliding distance 422 m. The COF is calculated using the data acquisition system of the tribological installation. A counter disk with a diameter of 140 mm and consisting of EN-31 steel hardened to 62 HRC (surface roughness: 1.6 Ra) was used for the wear experiments (Table 2).

**Table 2.** EN-31 steel composition.

| Element | C | Si | Mn | Cr | Si | Fe |
|---|---|---|---|---|---|---|
| Concentration, wt. % | 0.90–1.20 | 0.10–0.35 | 0.30–0.75 | 1.00–1.60 | 0.20 | rest |

## 3. Results and Discussion

### 3.1. Microstructure

Figure 3 displays SEM images with markers indicating different zones of the test specimen after wear tests (Figure 3a,b) and before wear test (Figure 3c,d). EDS analysis results are given in Tables 3 and 4, and in Figure 4. The SEM images were taken after the conducted tribological experiments with load of 50 N, linear speed of 1.0 m·s$^{-1}$ and sliding distance 422 m. The images in Figure 3 show that in the contact surface of all four test specimens, the dominant wear mechanism under the specified load and linear speed is abrasive wear. The abrasive wear mechanism is evidenced by the wear scar and wear direction on the contact zone, created by the sharp asperities during the tribological process. Due to the greater size of the pore walls and the larger area of surface interaction in Figure 3a, the abrasive wear mechanism is more evident and prevailing when compared with Figure 3b, where the area of surface interaction is reduced because of the larger pores. As a result of the continuous sliding under the applied load and linear speed of the interacting surfaces, a particular critical limit emerges in which surface deformations occur; because of the induced strain in the contact zone of the composite, the fatigue wear mechanism is evident very close to the pore walls of the contact zone of both specimens.

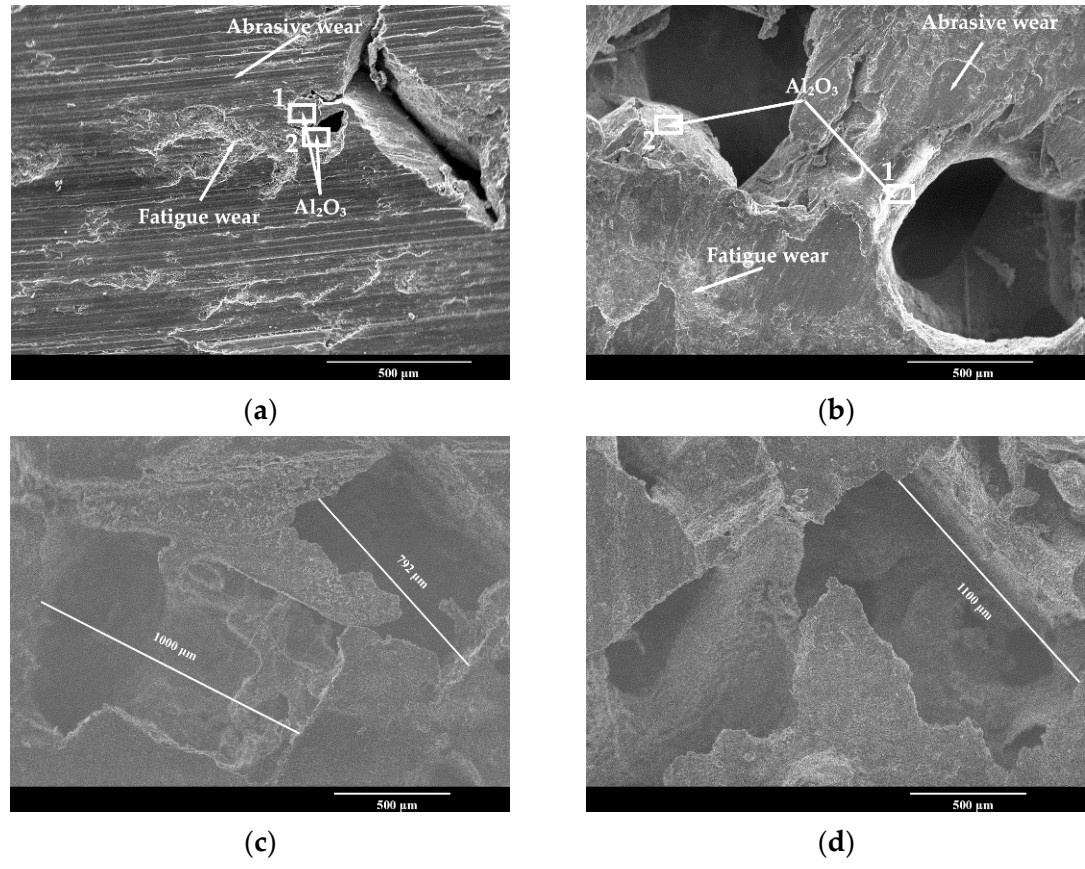

(**a**)　　　　　　　　　　　　　　　　(**b**)

(**c**)　　　　　　　　　　　　　　　　(**d**)

**Figure 3.** SEM images of AlSi10Mg-Al$_2$O$_3$ composites: (**a**) post-tribological tests of composite with pore size of 800 ÷ 1000 μm; (**b**) post-tribological tests of composite with pore size of 1000 ÷ 1200 μm; (**c**) before tribological tests of composite with pore size of 800 ÷ 1000 μm; (**d**) before tribological tests of composite with pore size of 1000 ÷ 1200 μm.

**Table 3.** EDS analysis of selected zones from Figure 3a of the wear surface of AlSi10Mg-Al$_2$O$_3$ with pore size of 800 ÷ 1000 μm, at a load of 50 N and linear speed of 1.0 m·s$^{-1}$, mass norm., %.

| No. Analysis | Al | Fe | O | Si | Mg | Mn | Other |
|:---:|:---:|:---:|:---:|:---:|:---:|:---:|:---:|
| 1 | 64.26 | 0.52 | 33.22 | – | – | 0.42 | rest |
| 2 | 61.06 | 5.87 | 32.20 | – | 0.05 | – | rest |

**Table 4.** EDS analysis of selected zones from Figure 3b of the wear surface of AlSi10Mg-Al$_2$O$_3$ with pore size of 1000 ÷ 1200 μm, at a load of 50 N, and linear speed of 1.0 m·s$^{-1}$, mass norm., %.

| No. Analysis | Al | Fe | O | Si | Mg | Mn | Other |
|:---:|:---:|:---:|:---:|:---:|:---:|:---:|:---:|
| 1 | 48.04 | 3.04 | 41.50 | 6.33 | 0.07 | – | rest |
| 2 | 53.42 | 11.23 | 29.15 | 4.47 | 0.18 | 0.23 | rest |

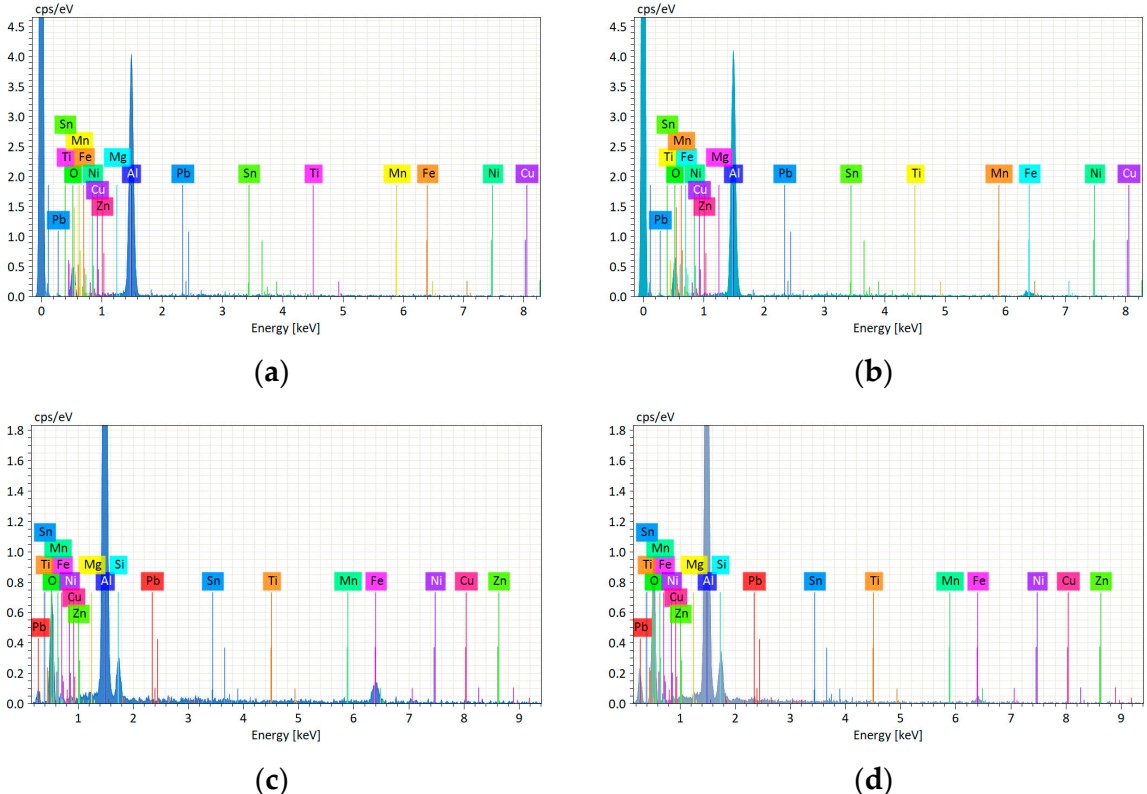

**Figure 4.** EDS spectra of the following skeleton specimens: (**a**) related to Table 3, analysis 1, AlSi10Mg-Al$_2$O$_3$ composite with pore size of 800 ÷ 1000 μm; (**b**) related to Table 3, analysis 2, AlSi10Mg-Al$_2$O$_3$ composite with pore size of 800 ÷ 1000 μm; (**c**) related to Table 4, analysis 1, Al alloy with pore size of 1000 ÷ 1200 μm; (**d**) related to Table 4, analysis 2, AlSi10Mg-Al$_2$O$_3$ composite with pore size of 1000 ÷ 1200 μm.

The marks in Figure 3a represent the composite skeleton with pore size 800 ÷ 1000 μm, highlighting two zones of EDS analysis, the results of which are present in Table 3 followed by their related EDS spectra in Figure 4a,b. It can be seen that in zone 1 and 2 there is a dominant peak of aluminum followed by oxygen, and small peaks of iron, which are more evident in zone 2. The above-mentioned arrangement of the chemical composition indicates the occurrence of the incorporation of alumina reinforcement into the pore walls of the AlSi10Mg-Al$_2$O$_3$ composite skeleton. The smaller peaks of iron content are a result of the detachment of asperities from the counter disk.

The composite skeleton with pore size 1000 ÷ 1200 μm is presented with the two marks in Figure 3b, each one highlighting different zones of EDS analysis, the results of which are shown in Table 4 followed by their EDS spectra shown in Figure 4c,d. It can be seen that in zone 1 and 2 there is a dominant peak of aluminum followed by oxygen, and small peaks of iron and silicon. The above-mentioned arrangement of the chemical composition suggests the occurrence of the incorporation of alumina reinforcement into the pore walls of the AlSi10Mg-Al$_2$O$_3$ composite. The smaller peaks of iron and silicon are as a result of the detachment of asperities from the counter disk and base alloy of the test specimen.

In order to add more evidence of the occurrence of reinforcement particles, in the light microscope image of Figure 5c three marks are presented, highlighting different zones of microhardness measurement of the reinforcement particles of AlSi10Mg-Al$_2$O$_3$ composite with pore size 800 ÷ 1000 μm. Again, in Figure 6c, the presence of reinforcement in the other composite skeleton with pore size 1000 ÷ 1200 μm is shown.

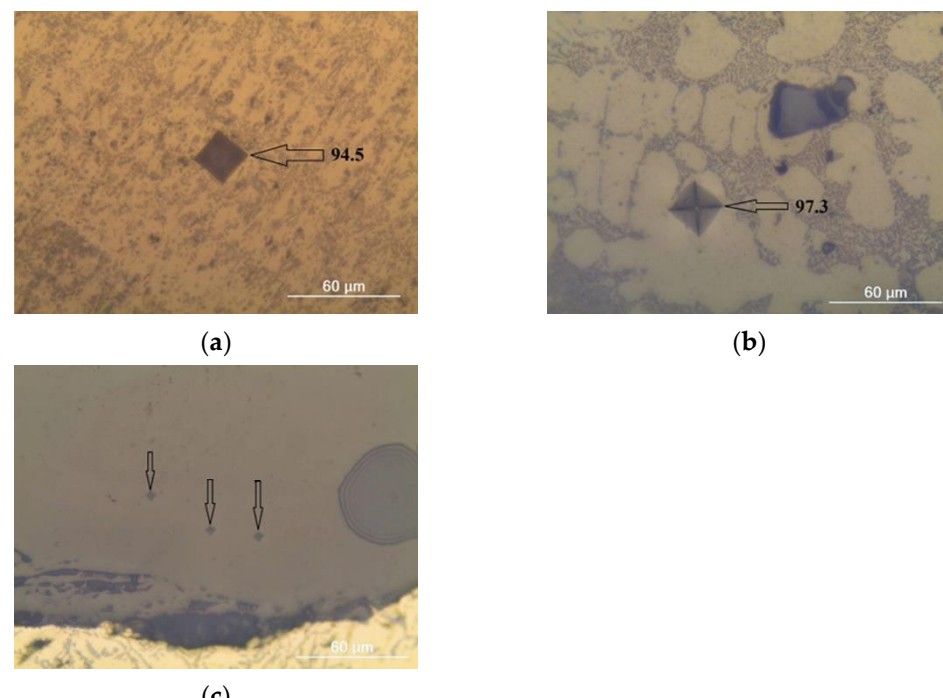

**Figure 5.** Light microscope images with micro-hardness measurement marks of skeleton specimens with pore size of 800 ÷ 1000 μm: (**a**) Al alloy matrix without reinforcement; (**b**) matrix of AlSi10Mg-Al$_2$O$_3$ composite, zone of matrix alloy; (**c**) f AlSi10Mg-Al$_2$O$_3$ composite, zone containing reinforcement. The size of the mark is much smaller compared to the marks presented in (**a**,**b**), due to measurement carried out on the reinforcement.

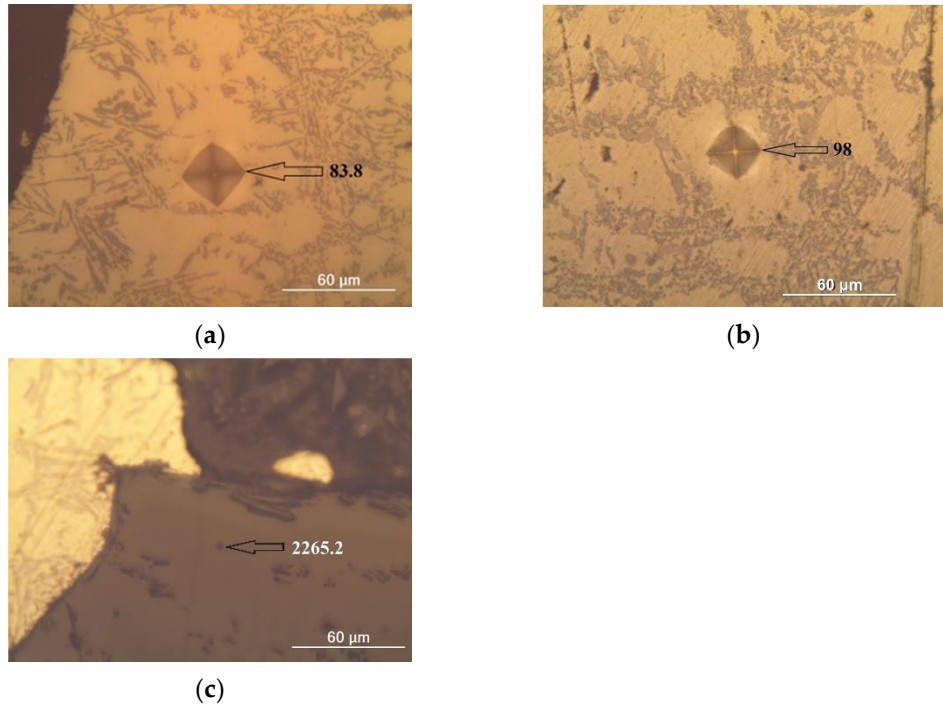

**Figure 6.** Light microscope images with micro-hardness measurement marks of skeleton specimens with a pore size of 1000 ÷ 1200 μm: (**a**) Al alloy matrix without reinforcement; (**b**) Matrix of AlSi10Mg-Al$_2$O$_3$ composite, zone of matrix alloy; (**c**) AlSi10Mg-Al$_2$O$_3$ composite zone containing reinforcement. The size of the mark is much smaller compared with the marks presented in (**a**,**b**) due to measurement carried out on the reinforcement.

### 3.2. Wear and Micro-Hardness Behavior

Average micro-hardness was measured at an applied force 0.05 kg·f, a time until attaining a specified load of 10 s and a time of retaining the load of 10 s on the cross-sections of the Al alloy and AlSi10Mg-Al$_2$O$_3$ composites with pore sizes of 800 ÷ 1000 μm and 1000 ÷ 1200 μm. The results are depicted in Figure 7a,b with supporting images of the light microscope in Figures 5a–c and 6a–c.

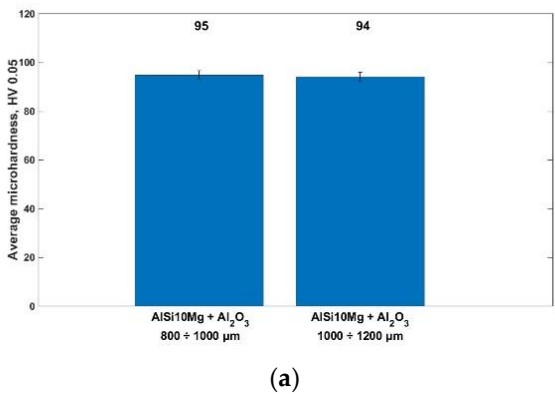
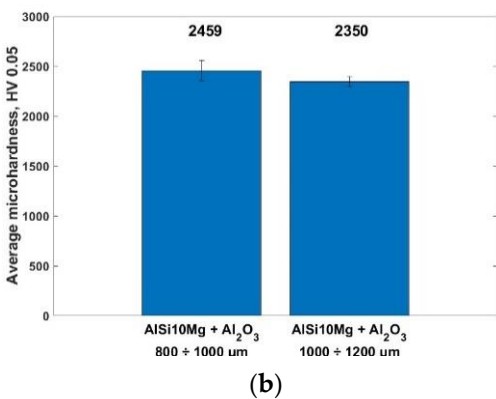

(**a**)　　　　　　　　　　　　　　　　　　　　　　　　　　　　(**b**)

**Figure 7.** Average micro-hardness at 0.05 kg·f. applied force, 10 s time until attaining specified load and 10 s time of retaining the load of the following zones in the skeleton specimen: (**a**) matrix of Al alloy and composite specimen with pore sizes of 800 ÷ 1200 μm and 1000 ÷ 1200 μm; (**b**) alumina particles of composite specimen with pore sizes of 800 ÷ 1200 μm and 1000 ÷ 1200 μm.

The results in Figure 7a indicate that at an applied force of 0.05 kg·f, a time until attaining specified load of 10 s and a time of retaining the load of 10 s, the Al alloy with smaller pores has a 10% improvement in hardness and the composite skeleton with the smaller pores has an insignificant improvement of 1% in micro-hardness.

In Figure 7b. the results show a 4% improvement in the hardness for the composite with the smaller pore size.

The results, with respect to the coefficient of friction at an applied load of 50 N, a linear speed of 1.0 m·s$^{-1}$ and a sliding distance 422 m at dry sliding conditions at room temperature, of the comparison of the base Al alloy skeleton vs. the composite skeleton with pore sizes of 800 ÷ 1200 μm and 1000 ÷ 1200 μm, are presented in Figure 8. The obtained results indicate that the introduction of the reinforcement in the composite skeleton with smaller pores (800 ÷ 1000 μm) decreases the COF by 4.2%, and in the composite skeleton with larger pores (1000 ÷ 1200 μm) a decrease of the COF by 3.2% is also noted. When we compare the two composite skeletons with respect to their pore sizes, the result shows that the COF remains practically unchanged (0.2%), and the same result occurs when we compare the two Al alloy skeletons with different pore sizes (0.8%).

The results, with respect to the mass wear at an applied load of 50 N, a linear speed of 1.0 m·s$^{-1}$ and a sliding distance of 422 m at dry sliding conditions at room temperature of the comparison of the base Al alloy skeleton vs. the composite skeleton with pore sizes of 800 ÷ 1200 μm and 1000 ÷ 1200 μm, are presented in Figure 9. The acquired results point out that the introduction of the reinforcement in the composite skeleton with smaller pores (800 ÷ 1000 μm) decreases the mass wear by 53.5%, and in the composite skeleton with larger pores (1000 ÷ 1200 μm) a decrease of the mass wear by 53.7% is also noted. When we compare the two composite skeletons with respect to their pore sizes, the results point out that the mass wear of the composite skeleton with bigger pores has a 6% decrease in mass wear, and when we compare the Al alloy skeletons with respect to their pore sizes, the results show that in the skeleton with the bigger pores, there is a decrease in the mass wear of 5.6%. The results from the comparison between the composites and the results from the comparison between the Al alloy skeletons are both within the limits of the measurement error resulting from the scattering of the results.

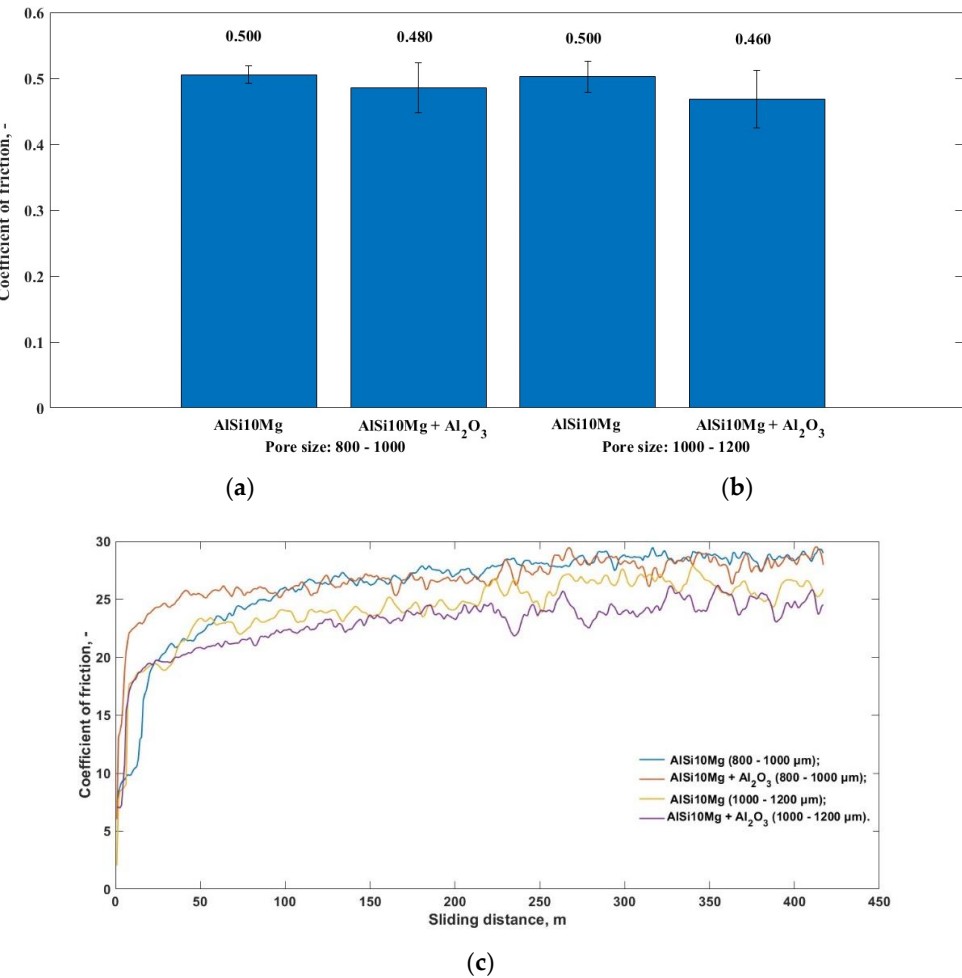

**Figure 8.** COF of Al alloy skeleton and AlSi10Mg-Al$_2$O$_3$ composite at 50 N load & 422 m sliding distance under dry friction conditions at room temperature with respect to pore sizes: (**a**) 800 ÷ 1000 μm; (**b**) 1000 ÷ 1200 μm; (**c**) COF vs. sliding distance test result.

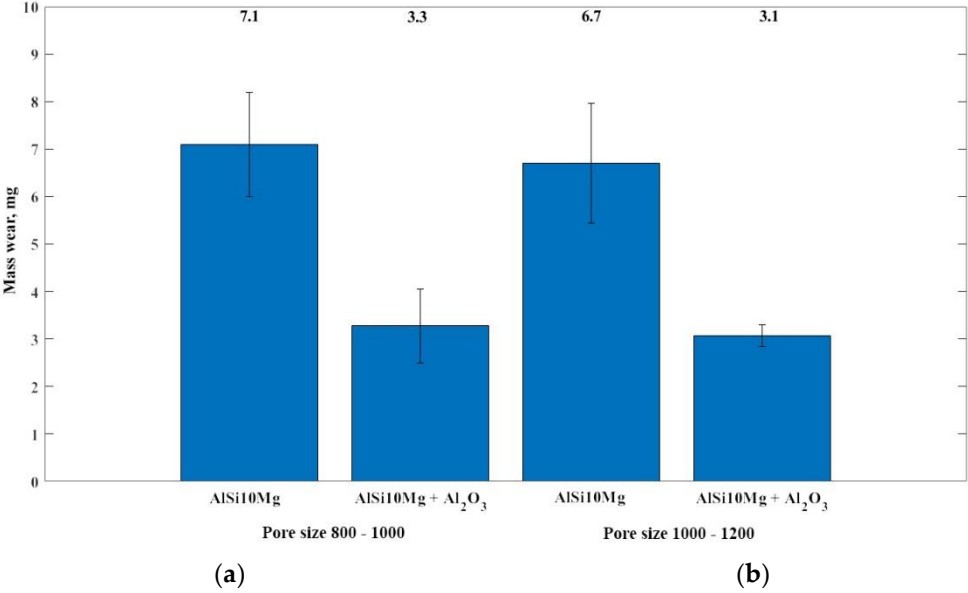

**Figure 9.** Mass wear of Al alloy skeleton and AlSi10Mg-Al$_2$O$_3$ composite at 50 N load & 422 m sliding distance under dry friction conditions at room temperature with respect to pore sizes: (**a**) 800 ÷ 1000 μm; (**b**) 1000 ÷ 1200 μm.

The main focus of the present research is the fabrication of an advanced open-cell AlSi10Mg-Al$_2$O$_3$ composite material and the investigation of the effect of tribological interaction parameters, such as mass wear and COF, under dry sliding conditions. Furthermore, structural and chemical characterization of the advanced composite material was performed.

By applying a load of 50 N with a sliding distance of 422 m and a linear velocity of 1.0 m·s$^{-1}$ against an EN-31 steel counter disk, the composite skeleton, with respect to both pore sizes (800 ÷ 1000 μm and 1000 ÷ 1200 μm), indicates great improvement in mass wear and a slight improvement in COF in comparison with the skeleton material. Due to the presence of alumina particles in pore walls, the latter are without visible signs of deformation and the inner pore surfaces are smooth and free of cracks thanks to the liquid-phase processing technology.

As a result of the inner pore surfaces being smooth and crack-free, with the increase of the pore size, the pore walls become thinner, and the composite skeleton as well as the skeleton material with the bigger pores suggest a slight improvement in their results in relation to the mass wear, and practically no improvement in the COF when compared with the composite skeleton and skeleton material with smaller pores. The reason for this is that the interactive surfaces have a decreased area of contact when compared with the specimen with smaller pores and a bigger surface area of contact.

## 4. Conclusions

In this study, a process to produce an advanced open-cell AlSi10Mg-Al$_2$O$_3$ composite skeleton was developed, and the effect of pore size on tribological parameters, such as the coefficient of friction and mass wear at 50 N load, 1.0 m·s$^{-1}$ linear speed and 422 m sliding distance under dry sliding conditions at room temperature, was investigated. The novelty of the present research is centered in the combination of the (1) low-density, fatigue strength, high load-carrying capacity, thermal conductivity, excellent corrosion resistance and overall low-price characteristics of the aluminum alloy with the (2) excellent wear and hardness behavior of the reinforcing Al$_2$O$_3$ particles, which might result in a potential practical application for the fabrication of sliding contact bearings.

Based on the performed tribological tests, a number of conclusions can be drawn, as follows:

- The AlSi10Mg-Al$_2$O$_3$ composite skeleton with pore size 800 ÷ 1000 μm decreases the COF 4.2% in comparison with the AlSi10Mg skeleton.
- The AlSi10Mg-Al$_2$O$_3$ composite skeleton with pore size 1000 ÷ 1200 μm decreases the COF 3.2% in comparison with the AlSi10Mg skeleton.
- Based on the above two facts, it can be concluded that the effect of the pore size in the range 800 ÷ 1200 μm does not affect the COF (1% difference, which is within the limits of the measurement error).
- The AlSi10Mg-Al$_2$O$_3$ composite skeleton with pore size 800 ÷ 1000 μm decreases the mass wear 53.5% in comparison with the AlSi10Mg skeleton.
- The AlSi10Mg-Al$_2$O$_3$ composite skeleton with pore size 1000 ÷ 1200 μm decreases the mass wear 53.7% in comparison with the AlSi10Mg skeleton.
- Based on the above two facts, it can be concluded that the effect of the pore size in the range 800 ÷ 1200 μm does not affect the mass wear (0.2% difference, which is within the limits of the measurement error).

**Author Contributions:** Conceptualization, M.K. and L.D.; methodology, all authors; formal analysis, all authors; investigation, M.K., V.P. and R.D.; writing—original draft preparation, M.K.; software, M.K., V.P. and R.D.; writing—review and editing M.K. and L.D.; visualization, M.K.; project administration, M.K.; funding acquisition, M.K. All authors have read and agreed to the published version of the manuscript.

**Funding:** This research was funded by the BULGARIAN NATIONAL SCIENCE FUND, Project КП-06-Н57/20 "Fabrication of new type of self-lubricating antifriction metal matrix composite materials with improved mechanical and tribological properties".

**Informed Consent Statement:** Not applicable.

**Data Availability Statement:** Not applicable.

**Acknowledgments:** This work is supported by the European Regional Development Fund within the OP Science and Education for Smart Growth 2014–2020, Project CoE National Center of Mechatronics and Clean Technologies, BG05M2OP001-1.001-0008.

**Conflicts of Interest:** The authors declare no conflict of interest.

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
