# Peer review of "Production and Tribological Characterization of Advanced Open-Cell AlSi10Mg-Al2O3 Composites"

_metals, doi:10.3390/met13010131_

Round 1

Reviewer 1 Report

Dear Authors,

The paper presents research on the fabrication and investigation of the effect of tribological interaction parameters under dry sliding conditions, such as mass wear and COF of an open-cell AlSi10Mg- Al2O3 composite material. Structural, and chemical characterization of the composite material were performed. This study is a continuation of Authors’ previous studies.

The manuscript is well structured, but needs some corrections and additions, for example:

Lines 135: Specifying three significant places for linear velocity (1.004 ms-1) is an exaggeration - you can probably round up to 1.0ms-1.

Lines 173-177: XRD analysis can be performed to confirm the presence of an Al2O3 reinforcing phase with a crystalline structure.

Lines 197-198 and Figure 7: Showing a 1% or 4% change in hardness is probably within the measurement error. Figure 7 does not indicate hardness measurement errors.

Figure 8: Given three significant places (0.506, etc.) for average COF values, this is too accurate given the measurement error.

Lines 204-215 and Figure 9: Demonstration of differences in the magnitude of the mass wear depending on the size of the composite pore is within the limits of the measurement error resulting from the scattering of the results. The influence of the reinforcing phase is clearly visible.

Lines 248-264: Conclusions are based on test results, where the differences in results were within the limits of measurement errors - so they create a lot of uncertainty.

The manuscript can be accepted after the necessary additions and corrections.

 Kind regards,

Author Response

Dear Reviewer,

Thank you for your kind comments and suggestions about our manuscript with the title “Production and Tribological Characterization of Advanced Open-Cell AlSi10Mg-Al2O3 Composites”. It would help us to improve the quality of our manuscript. We have inspected the whole manuscript and tried our best to revise carefully according to your comments in our revised version of the manuscript.

  1. Lines 135: Specifying three significant places for linear velocity (1.004 ms-1) is an exaggeration - you can probably round up to 1.0ms-1.

Response: As suggested by the reviewer (1.004 ms-1) is rounded to (1.0 ms-1) in the manuscript.

  1. Lines 173-177: XRD analysis can be performed to confirm the presence of an Al2O3 reinforcing phase with a crystalline structure.

Response: We appreciate the reviewer’s suggestion, however, it could be done if we did not have a preliminary analysis, but since we know that in addition to the aluminum alloy, we have introduced Al2O3 and there is no reason to expect that the visible phase could be anything different.

  1. Lines 197-198 and Figure 7: Showing a 1% or 4% change in hardness is probably within the measurement error. Figure 7 does not indicate hardness measurement errors.

Response: As recommended by the reviewer the hardness measurement error bars are present.

  1. Figure 8: Given three significant places (0.506, etc.) for average COF values, this is too accurate given the measurement error.

Response: As suggested by the reviewer the COF values are rounded.

  1. Lines 204-215 and Figure 9: Demonstration of differences in the magnitude of the mass wear depending on the size of the composite pore is within the limits of the measurement error resulting from the scattering of the results. The influence of the reinforcing phase is clearly visible.

Response: As proposed by the reviewer the text in lines 204-215 commenting Fig. 9 are edited and made clearer.

  1. Lines 248-264: Conclusions are based on test results, where the differences in results were within the limits of measurement errors - so they create a lot of uncertainty.

Response: As suggested by the reviewer the text in lines 248-264 is edited and made clearer.

Once again, we appreciate for your comments and suggestions.

Reviewer 2 Report

In this work, AlSi10Mg-Al2O3 composite with open-cell porous was fabricated by replication of NaCl space-holders. The tribological behavior of fabricated composites with different pore sizes was investigated under dry sliding conditions at room temperature via the pin-on-disk method. It contains interesting results, however, also missing important information. It needs substantial revision.

1.      Introduction: Why interconnected (open-cell) aluminum metal matrix composites have been served as a vital component of many functional and structural engineering applications? Please give the necessary explanations.

2.      The size of Al2O3 is between 300 and 400 μm, why use such large size aluminum oxide? It is also strange that in Fig.6, the size of Al2O3 seems very small.

3.      The scale bar in all SEM pictures is difficult to see clearly.

4.      Please show the microstructure of the fabricated composites before wearing test.

5.      If you discuss the effect of pore size on wear resistance property, then analysis on pore size, as well as pore morphology and porosity should be given and compared.

6.      The obtained result in this work cannot support the conclusion that the effect of the pore size to the mass wear for the skeleton material is negligible, as the two selected pore size are lack of representative.

7.      The language expression needs to be carefully checked and revised.

Author Response

Reviewer 2

Dear Reviewer,

Thank you for your kind comments and suggestions about our manuscript with the title “Production and Tribological Characterization of Advanced Open-Cell AlSi10Mg-Al2O3 Composites”. It would help us to improve the quality of our manuscript. We have reviewed the whole manuscript and tried our best to revise carefully according to your suggestions in our revised version of the manuscript.

  1. Introduction: Why interconnected (open-cell) aluminum metal matrix composites have been served as a vital component of many functional and structural engineering applications? Please give the necessary explanations.

Response: As suggested by the reviewer the necessary explanations have been written in the manuscript.

  1. The size of Al2O3 is between 300 and 400 μm, why use such large size aluminum oxide? It is also strange that in Fig.6, the size of Al2O3 seems very small.

Response: We appreciate the reviewer’s recommendation, however, this study is a continuation of our previous studies. As is stated in the aim of current research part from the introduction section, in our previous study we use Al2O3 particles in the range of 0 – 30 μm.

As recommended by the reviewer, the highlights on the images and the captions of Fig. 6 are edited in order to be clear for the readers.

  1. The scale bar in all SEM pictures is difficult to see clearly.

Response: The scale bar in all SEM images is now clear and visible.

  1. Please show the microstructure of the fabricated composites before wearing test.

Response: The microstructure and the pore size of the composite materials before conducting the wear tests are calculated by the SEM image processing software and presented in Figure 3 (c) and (d).

  1. If you discuss the effect of pore size on wear resistance property, then analysis on pore size, as well as pore morphology and porosity should be given and compared.

Response: With respect to the reviewers comment the morphology of all test samples is the same, since the used salt particles have the same geometric parameters. For this reason, all samples are with the same pore morphology and there is no base to compare the effect of pore morphology.

  1. The obtained result in this work cannot support the conclusion that the effect of the pore size to the mass wear for the skeleton material is negligible, as the two selected pore size are lack of representative.

Response: We appreciate the reviewer’s suggestion, however, as it was written in the response to question 5, the morphology of all test samples is the same, since the used salt particles have the same geometric parameters. For this reason, all samples are with the same pore morphology and there is no base to compare the effect of pore morphology.

  1. The language expression needs to be carefully checked and revised.

Response: The language expression is checked and revised.

Once again, thank you very much for your comments and suggestions.

Reviewer 3 Report

The author investigated the fabrication and tribological behavior of open-cell AlSi10Mg composite reinforced by Al2O3 particles using NaCl as space-holders. I have some comments and suggestions as bellow:

1. The porosity of the porous composite should be calculated and presented

2. Pore size of composite should be calculated using SEM or optical image using the statical values from image analysis software (i.e ImageJ, Saram Soft, etc…)

3. Distribution of Al2O3 inside AlSi10Mg matrix should be pointed out. The presented results (Figure 3) cannot be accepted.

4. Microhardness of the AlSi10Mg/Al2O3composite is nearly the same with that of AlSi10Mg alloy for pore size of 800-1000 µm. Please explain why there is no effect of Al2O3 on the hardness of the composite in this case.

5. Please include the figure of the friction coefficient test result versus distance and calculate the wear rate of the composite with and without Al2O3.  

6. How about the compressive strength of AlSi10Mg/Al2O3 composite?

Author Response

Reviewer 3

Dear Reviewer,

Thank you for your kind comments and suggestions about our manuscript with the title “Production and Tribological Characterization of Advanced Open-Cell AlSi10Mg-Al2O3 Composites”. It would help us to improve the quality of our manuscript. We have inspected the whole manuscript and tried our best to revise carefully according to your comments in our revised version of the manuscript.

  1. The porosity of the porous composite should be calculated and presented

Response: As suggested by the reviewer the porosity of the porous composite and porous material are calculated and included at the end of the subsection “Production Method and Materials”.

  1. Pore size of composite should be calculated using SEM or optical image using the statical values from image analysis software (i.e ImageJ, Saram Soft, etc…)

Response: As suggested by the reviewer the pore size of the composite materials are calculated by the SEM image processing software and presented in Figure 3 (c) and (d).

  1. Distribution of Al2O3 inside AlSi10Mg matrix should be pointed out. The presented results (Figure 3) cannot be accepted.

Response: We understand the reviewer’s recommendation, however, it could be done if we did not have a preliminary analysis, but since we know that in addition to the aluminum alloy, we have introduced Al2O3 and there is no reason to expect that the visible phase could be anything different.

  1. Microhardness of the AlSi10Mg/Al2O3 composite is nearly the same with that of AlSi10Mg alloy for pore size of 800-1000 µm. Please explain why there is no effect of Al2O3 on the hardness of the composite in this case.

Response: We understand the reviewer’s comment, for which reason Fig. 7 is edited and made more explicit.

  1. Please include the figure of the friction coefficient test result versus distance and calculate the wear rate of the composite with and without Al2O3.

Response: The COF vs sliding distance is included in Figure 8 (c).

  1. How about the compressive strength of AlSi10Mg/Al2O3 composite?

Response: Of course, compressive behaviour is very important for practical application, but this subject needs specific study which will be conducted and discussed in our future manuscript.  

 We appreciate your comments and suggestions.

Round 2

Reviewer 1 Report

Dear Authors,

I agree with the Authors' responses and noted that they have taken into account my comments and made corrections to the manuscript.

The manuscript in this form can be accepted for publication. Although I still think that the main conclusions are based on the results of the study, where the differences in the results were within the limits of measurement errors and therefore raise uncertainty.

Kind regards,

Author Response

We would like to thank the reviewer for taking the necessary time and effort to review the manuscript. We sincerely appreciate all the valuable comments and suggestions, which helped us in improving the quality of the manuscript.

Reviewer 2 Report

The revised manuscript was improved a lot. But there are still some points need to be clarified.  (1) How you determined the porosity of the fabricated two composites? (2) What is the volume content of Al2O3 in the two fabricated composites? Microstructure pictures which can distinguish the distribution of Al2O3 in the matrix should be presented.  (3) In Fig. 7, why the micro-hardness of Al2O3 reinforcement in the two composites is different?

Author Response

Reviewer 2

Dear Reviewer,

Thank you for your kind comments and suggestions about our manuscript with the title “Production and Tribological Characterization of Advanced Open-Cell AlSi10Mg-Al2O3 Composites”. It would help us to improve the quality of our manuscript. We have reviewed the whole manuscript and tried our best to revise it carefully according to your suggestions in our revised version of the manuscript.

  1. How you determined the porosity of the fabricated two composites?

Response: The porosity of the fabricated two composites was estimated by the following equation:

... equation is presented in the attached docx file

Where P is the porosity in percentage, ρth is the density of nominally nonporous composite and ρc is the measured density of the composite material.

Since the equation is well-known, simple, and basic it is not included in the manuscript.

  1. What is the volume content of Al2O3 in the two fabricated composites? Microstructure pictures which can distinguish the distribution of Al2O3 in the matrix should be presented.

Response: In the first part of the materials and methods section we have included in the text that the ceramic particles are 5 wt. % of the composite materials. We consider that there is no reason to expect any changes in composite composition during samples preparation and because the volume content of Al2O3 in the AlSi10Mg + Al2O3 composite is known (5 wt. %) we have not presented microstructure pictures.

  1. In Fig. 7, why the micro-hardness of Al2O3 reinforcement in the two composites is different?

Response: Since the difference in the micro-hardness results of Al2O3 reinforcement in the two composites is less than 5 % (2459 and 2350) we can consider that the reason for this small difference is due to the placement of the indenter and also the angle of the positioning of the ceramic particle.

Once again, thank you very much for your comments and suggestions.

Reviewer 3 Report

The manuscript can be accepted in its current form.

Author Response

(The authors gave the same response as above.)
